# Inversion of Different Cultivated Soil Types' Salinity Using Hyperspectral Data and Machine Learning

**Pingping Jia** [1,2,†], **Junhua Zhang** [3,4,†], **Wei He** [1], **Ding Yuan** [1], **Yi Hu** [1], **Kazem Zamanian** [1,5], **Keli Jia** [2] **and Xiaoning Zhao** [1,*]

1 School of Geographical Sciences, Nanjing University of Information Science and Technology, Nanjing 210044, China
2 School of Geography and Planning, Ningxia University, Yinchuan 750021, China
3 School of Ecology and Environment, Ningxia University, Yinchuan 750021, China
4 Breeding Base for State Key Laboratory of Land Degradation and Ecological Restoration in Northwestern China, Ningxia University, Yinchuan 750021, China
5 Institute of Soil Science, Leibniz University of Hannover, 30419 Hannover, Germany
* Correspondence: zhaoxiaoning@nuist.edu.cn
† These authors contributed equally to this work.

**Abstract:** Soil salinization is one of the main causes of global desertification and soil degradation. Although previous studies have investigated the hyperspectral inversion of soil salinity using machine learning, only a few have been based on soil types. Moreover, agricultural fields can be improved based on the accurate estimation of the soil salinity, according to the soil type. We collected field data relating to six salinized soils, Haplic Solonchaks (HSK), Stagnic Solonchaks (SSK), Calcic Sonlonchaks (CSK), Fluvic Solonchaks (FSK), Haplic Sonlontzs (HSN), and Takyr Solonetzs (TSN), in the Hetao Plain of the upper reaches of the Yellow River, and measured the in situ hyperspectral, pH, and electrical conductivity (EC) values of a total of 231 soil samples. The two-dimensional spectral index, topographic factors, climate factors, and soil texture were considered. Several models were used for the inversion of the saline soil types: partial least squares regression (PLSR), random forest (RF), extremely randomized trees (ERT), and ridge regression (RR). The spectral curves of the six salinized soil types were similar, but their reflectance sizes were different. The degree of salinization did not change according to the spectral reflectance of the soil types, and the related properties were inconsistent. The Pearson's correlation coefficient (PCC) between the two-dimensional spectral index and the EC was much greater than that between the reflectance and EC in the original band. In the two-dimensional index, the PCC of the HSK-NDI was the largest (0.97), whereas in the original band, the PCC of the $SSK_{400\ nm}$ was the largest (0.70). The two-dimensional spectral index (NDI, RI, and DI) and the characteristic bands were the most selected variables in the six salinized soil types, based on the variable projection importance analysis (VIP). The best inversion model for the HSK and FSK was the RF, whereas the best inversion model for the CSK, SSK, HSN, and TSN was the ERT, and the CSK-ERT had the best performance ($R^2 = 0.99$, RMSE = 0.18, and RPIQ = 6.38). This study provides a reference for distinguishing various salinization types using hyperspectral reflectance and provides a foundation for the accurate monitoring of salinized soil via multispectral remote sensing.

**Keywords:** soil electrical conductivity; variable projection importance; Hetao Plain; salinization; soil degradation; soil quality

## 1. Introduction

Salinization causes a decline in soil fertility, deteriorates the ecological environment, and is one of the main factors restricting the sustainable development of agricultural production and the ecological environment [1–3]. Saline soils cover nearly one billion hectares of land in more than 100 countries [4], and it is estimated that 50% of the globe's arable land will be salinized by 2050 [5]. This situation has hindered the realization of sustainable development goals across the world, and salinization has become a worldwide ecological problem [6,7]. The management and utilization of salinized soil is very important for regional food production, ecological security, and sustainable agricultural development.

In particular, saline soils account for about 10% of China's national land area [8]. The Hetao Plain in the upper reaches of the Yellow River is an important location for grain and sugar production, for both China and the rest of the world. However, long-term irrigation by the Yellow River, coupled with the high salt content of the parent material, fluctuations in the groundwater level, droughts, and strong evaporation have increased the area of saline–alkaline land and the degree of salinization in the region [9]. Thus, salinization has become a major obstacle for appropriate land use in Hetao Plain [10], and the study of the salinization rates and characteristics is crucial for the effective prevention of salinization and improvement in regional productivity.

Different soil types determine the management measures for crops and agriculture. Many scholars have carried out research addressing the salinization threat to agriculture and salinization distribution, as well as establishing inversion models [11,12]; they have proposed the implementation of targeted techniques (geostatistical methods) for determining spatial variations in soil salinity [13]. The electrical conductivity (EC) of soil is closely related to the degree of salinity and is widely used in salinization-related studies [14]. For different types of saline–alkaline soil, establishing a correlation equation between the EC and the degree of soil salinity and alkalinity and combining the spatial distribution of the salinity and alkalinity with agricultural management, such as irrigation and fertilization regimes, have attracted increasing attention [15].

Remote sensing has been successfully applied for the monitoring of EC, ranging from hyperspectral to multispectral analysis [16–18]. Hyperspectral remote sensing technology is an important method for monitoring the physical and chemical properties of soil because of its strong dynamics, high resolution, and continuous band. The field-measured hyperspectrum has been proven to retrieve soil salinity data with high accuracy [19]. However, soil salinization inversion studies in different regions vary in terms of the best model, spectral processing method, and variable screening method [20]. In addition, previous salinity inversion models have only used one soil type with a specific degree of salinity together with a specific salinization mechanism [21,22]. As a result, there is a lack of clarity on the variation in the spectral characteristics among saline soils with different degrees of salinity and salinization mechanisms, as well as the suitability of specific models for each soil type.

Soil has a high degree of spatial heterogeneity, and the soil moisture, texture, and depth will all affect the soil salinity. Therefore, the addition of auxiliary variables improves the accuracy of spectral prediction models [23,24]. However, to date, few scholars have added, for example, soil texture and depth as covariates to the models [23]. Moreover, the studies that have used modeling to investigate the response of environmental variables to soil salinization [25] did not consider whether these variables maintained consistent high efficiency under small-scale or under relatively uniform conditions.

For an accurate estimation of salinization, we selected six cultivated saline soil types for this research. Our main objectives were to (1) explore the response of spectra to various soil types with specific properties, EC values, and salinization mechanisms; (2) study the feasibility of the measured hyperspectrum to estimate soil EC values; (3) verify the inversion accuracy of different models in relation to soil type; and (4) propose a method for the construction of spectral quantitative inversion models based on soil type and its applicability for saline soils with various degrees of salinity and a wide range of properties.

## 2. Materials and Methods

### 2.1. Study Area

Hetao Plain (40°10′~41°20′N, 106°10′~112°15′E) is located in the center of the Inner Mongolia Plateau and along the Yellow River, between the Inner Mongolia Autonomous Region and the Ningxia Autonomous Region, and has a total area of about 28,729 km² [26]. It is an important grain producing area in northwest China. However, it suffers from sparse precipitation and strong evaporation (Figure 1d). The Yellow River Diversion Project has been flooded for a long time; despite this, the drainage in this area is poor, which has resulted in shallow groundwater, serious secondary salinization of soil, and an extremely fragile ecology. Under such conditions, the saline soil area of the Ningxia Yellow River Diversion irrigation area in Hetao Plain was 2.2755 million mu [27].

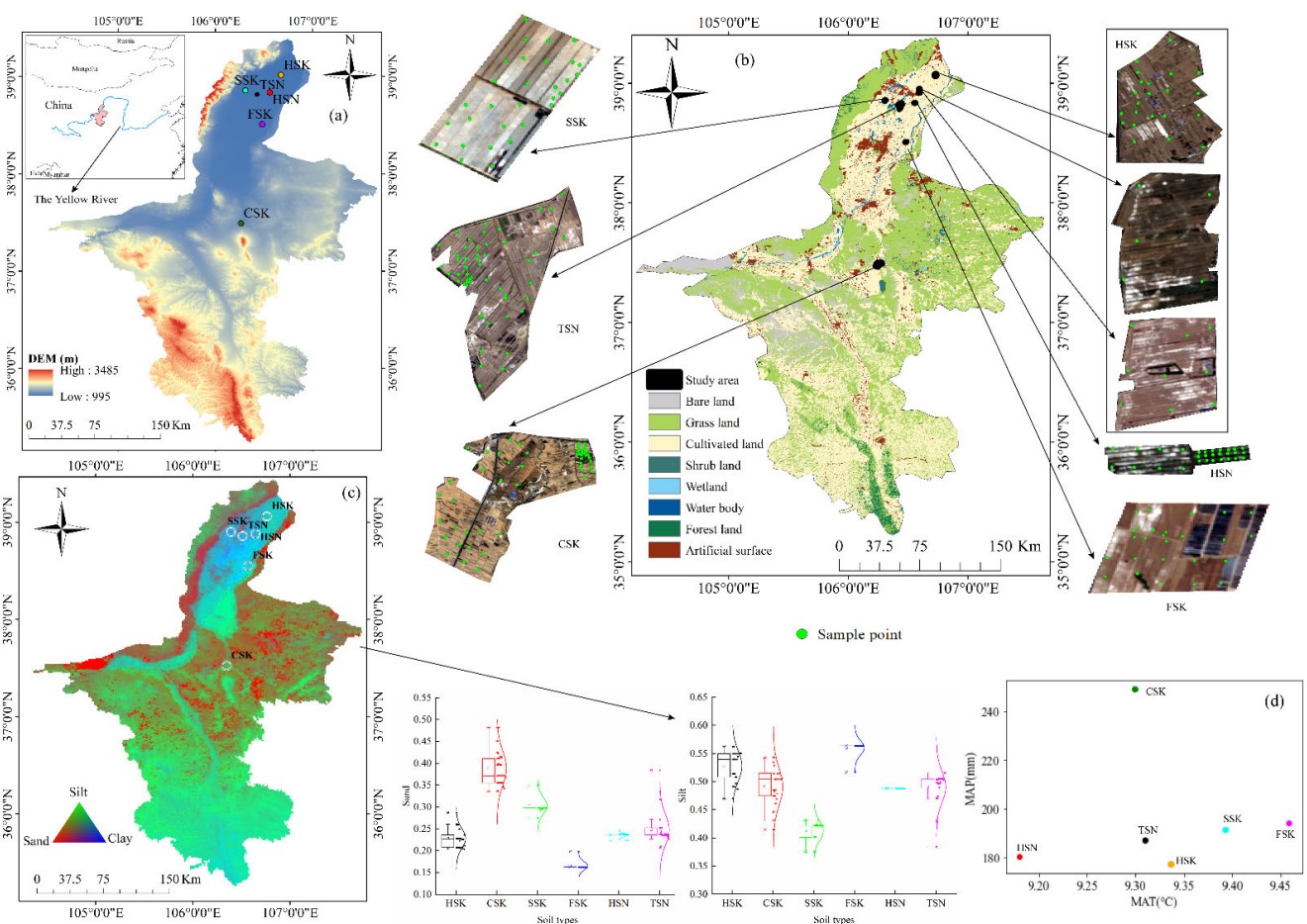

**Figure 1.** Location of the sampling sites in the study area. (**a**) Digital elevation model (DEM) of Ningxia autonomous in China; (**b**) distribution of sampling points; (**c**) variations in soil textures in the study area; (**d**) mean annual precipitation (MAP) and mean annual temperature (MAT) of the sampling sites. HSK is Haplic Solonchaks, SSK is Stagnic Solonchaks, CSK is Calcic Sonlonchaks, FSK is Fluvic Solonchaks, HSN is Haplic Sonlontzs, and TSN is Takyr Solonetzs.

We considered factors such as the soil surface characteristics, pH conditions, soil types, and land use patterns in the study area. Six typical saline–alkaline soil types in the upper reaches of the Yellow River (Ningxia section) were selected, including Haplic Solonchaks (HSK), Stagnic Solonchaks (SSK), Takyr Solonetzs (TSN), Haplic Sonlontzs (HSN), Fluvic Solonchaks (FSK), and Calcic Sonlonchaks (CSK). The HSK is found close to the diversion channel, where because of lateral seepage and blockage of the drainage ditch, the ground water level rises, and the salinization is aggravated. The SSK area is

located on low-lying terrain, with poor drainage and low water resource utilization efficiency, which has led to soil salinization. Sodium carbonate and sodium bicarbonate are the main salts in the TSN, and the $CO_3^{2-}$ and $HCO_3^-$ content accounts for more than 80% of the total anions [28]. The surface of the TSN comprises salt crusts with gray–white turtle cracks that are about 1 cm thick [29]. The soil clay content in the HSN is high (Figure 1c), and the salinization is aggravated by the non-standardized agricultural cultivation. The FSK is a low terrain close to the Yellow River, with poor drainage, and the soil contains minerals from the Yellow River. The parent material of the CSK has a high salt content, and there is a layer of impermeable calcium deposits in the soil profile [30].

### 2.2. Data Sources

### 2.2.1. Hyperspectral Data Acquisition and Preprocessing

Soil spectra were measured after the harvest at each sampling site. The CSK was collected on 10 to 11 March 2022, the SAS was collected on 30 March 2022, and the other samples were collected from 31 March to 10 April 2022. The soils were sampled using a grid method (Figure 1). Soil spectroscopy was conducted at each sampling site using the Analytical Spectral Devices (ASD) FieldSpec4 spectrometer (Analytical Spectral Devices, Inc., US). The detection band was 350–2500 nm, and the resampling interval was 1 nm. The resolution for 350–1000 nm was 3.5 nm, for 1000–1500 nm it was 10 nm, and for 1500–2100 nm it was 7 nm. The time of measurement was 10:00–14:00 on a sunny day, the spectrometer was facing vertically downward, and the probe was about 30 cm perpendicular to the surface. Standard whiteboard correction was performed before each collection, each sample point was measured five times, and the average value was taken as the spectral reflection value of the sample point.

Preprocessing: (1) The abnormal spectral curve removal, breakpoint correction, and the measured field spectral data were reperformed using in ViewSpec Pro software (A click view graph was used to delete the abnormal curve. The ASD spectrometer has three sensors, which have varying responsivity under different environmental function temperatures and warm-up times. Different optical fibers collect spectra of samples at different locations, and the splice correction function in the software was required to correct the data). (2) To eliminate instrument noise and environmental background interference, the edge bands with excessive noise (350–399 and 2401–2500 nm) were removed. (3) The Savitzky–Golay (polynomial order 2, number of smoothing points 9) method was used to smooth and denoise the 400–2400 nm data. (4) The spectral data of 400–2400 nm were resampled at 10 nm intervals, and 201 bands were obtained.

The bands with the largest Pearson's correlation coefficient (PCC) between the EC and the bands of blue (455–492 nm), green (492–577 nm), red (622–770 nm), near-infrared (770–1050 nm), swir1 (1500–1750 nm), and swir2 (2080–2350 nm) were selected as band modeling factors for use in the subsequent modeling.

### 2.2.2. Soil Sample Collection and Preprocessing

After spectral collection, the unmixed soil samples from 0 to 20 cm were collected at the same points using a soil drill and stored in sealed bags. A handheld GPS was used to record the longitude and latitude of each point, the sampling date, and the corresponding number of the soil sample and spectrum, together with information regarding the surface salt aggregation and land use pattern. The soil moisture content (%) was determined using the oven-drying and weighing method. Impurities such as gravels and weed remains were removed from the collected soil samples. After natural air-drying and grinding, the 1:5 soil: water mixture was prepared in order to measure the EC using an EC meter (FE38-Standard, Mettler Toledo, Switzerland). A total of 231 soil samples were collected: 53 from the CSK, 26 from the FSK, 30 from the SSK, 37 from the HSK, 29 from the HSN, and 56 from the TSN. According to the definition of Brady and Weil [31], the soils were grouped into five salinity levels: non-saline (EC < 0.4 ms/cm), slightly saline (0.4 ≤ EC < 0.8 ms/cm),

moderately saline ($0.8 \leq EC < 1.6$ ms/cm), strongly saline ($1.6 \leq EC < 2.4$ ms/cm), and extremely saline ($EC \geq 2.4$ ms/cm).

### 2.2.3. Environmental Variables

The climatic data were obtained from the National Meteorological Information Center. (http://data.cma.cn (accessed on 20 September 2020)). The mean annual temperature (MAT) and mean annual precipitation (MAP) in Ningxia for the last 40 years (1978–2018) were determined based on records from 10 stations. The preprocessed meteorological data were then used in ArcGIS 10.4 to generate the MAP and MAT of the sampling points in Ningxia using inverse distance weighted (IDW) in Raster Interpolation module.

Terrain data with a spatial resolution of 12.5 m were obtained from NASA's Alaska Satellite Facilities Division (http://search.asf.alaska.edu/#/ (accessed on 15 July 2022)). In ArcGIS 10.4, the "Extract Multivalues to Points" tool in Spatial Analyst Tools was used to extract the digital elevation model (DEM) of each sampling point, as well as the slope degree, slope aspect, plane curvature, profile curvature, and the topographic wetness index (TWI).

The soil texture data, i.e., the sand, silt, and clay content in the surface layer (g/kg) with a spatial resolution of 1 km, were collected from the basic attribute dataset of China's high-resolution National Soil Information Network provided by the National Earth System Science Data Center (http://www.geodata.cn (accessed on 16 October 2021)), and the soil depth to bedrock data were collected from [32].

### 2.3. Selection of the Optimal Spectral Index for Estimating Soil Salinity

The optimal band combination algorithm is able to fully consider the correlation information between bands and reduce interference from irrelevant wavelengths. In addition, the numerical two-dimensional contour map of the correlation between spectral index and salinity can provide comprehensive information regarding the ability of two different wavelength combinations to predict soil properties [33]. It has been pointed out that the second derivative of spectral reflectance is the best way to calculate the two-dimensional salinity index [34]; therefore, 2D correlation maps after the second derivative of reflectance were used to determine the relationship between the difference index (DI), the ratio index (RI), the normalized index (NDI), and the soil EC (Table 1).

**Table 1.** Reference overview of the studies on spectral indices and formulas.

| Acronym | Spectral Indices | Formula | Reference |
|---|---|---|---|
| DI | Difference Index | $R_i - R_j$ | [35] |
| RI | Ratio Index | $\dfrac{R_i}{R_j}$ | [35] |
| NDI | Normalized Index | $\dfrac{R_i - R_j}{R_i + R_j}$ | [36] |

$R_i$ and $R_j$ in the formula belong to the reflectance after the second derivative of any two wavelengths between 400 and 2400 nm, and $R_i \neq R_j$. For each spectral index, the wavelength combination with the largest correlation with soil EC was extracted and deemed to be the optimal band combination.

### 2.4. Method

#### 2.4.1. Features Selection

The variable projection importance analysis (*VIP*) is a variable screening method based on partial least squares regression (PLSR) [37]. For a given independent variable, the *VIP* value not only represents the effect of the independent variable on the dependent variable but also takes into account the indirect influence of other independent variables on the dependent variable. The calculation of the *VIP* is:

$$VIP_j = \sqrt{\frac{p * \sum_{f=1}^{F} SSY_f * W_{jf}^{2}}{SSY_{total} * F}}$$

where $p$ is the number of independent variables, $F$ is the total number of principal components, $f$ is the principal component, $SSY_f$ is the sum of squared variances explained by the $f$ principal component, $SSY_{total}$ is the sum of squares of dependent variables, and $W_{jf}^{2}$ gives the importance of the j variable in the $f$ principal component. The larger the value of $VIP_j$, the stronger the explanatory power of the independent variable to the dependent variable. When the $VIP$ value of the independent variable is greater than 1, the independent variable is judged as an important independent variable [38].

### 2.4.2. Modeling Method and Model Evaluation Index

The modeling methods were PLSR, random forest (RF), extremely randomized trees (ERT), and ridge regression (RR). The prediction performance of the model was evaluated using the fivefold cross validation method. The stability and prediction accuracy of the model were evaluated using the determination coefficient ($R^2$), root mean square error ($RMSE$), and the ratio of performance to interquartile distance ($RPIQ$). For the model calculation process, the GridSearchCV method was selected for hyperparameter tuning, i.e., the main parameters were found using the grid search method, and the other parameters were the default values of the Scikit–Learn tool kit. GridSearchCV ensured that the parameter with the highest precision could be found within the specified parameter range, where the PLSR search space was param_grid = {'n_components': range (1,20)}, RF: 'n_estimators': range (2,50,1), 'max_features': (2, 4, 6, 8), 'max_depth': range (2, 15, 2); ERT: 'n_estimators': range (1, 30, 1), 'max_depth': range (2, 15, 1), and RR: alphas = (0.01, 0.1, 0.5, 1, 5, 7, 10, 30,100).

$$R^2 = \frac{\sum_{i=1}^{n}(\hat{y}_i - \bar{y}_i)^2}{\sum_{i=1}^{n}(y_i - \bar{y}_i)^2} \tag{1}$$

where $y_i$ and $\hat{y}_i$ is the observed value and predicted value of the test sample, respectively, $\bar{y}_i$ is the average of sample observations, and $n$ is the number of predicted samples.

$$\text{RMSE} = \sqrt{\frac{1}{N}\sum_{i=1}^{n}(y_i - \hat{y}_i)} \tag{2}$$

where $\hat{y}_i$ is the predicted value of the sample, and $y_i$ is the measured value.

$$\text{RPIQ} = \frac{IQ}{RMSE} \tag{3}$$

where $IQ$ is the difference between the third quartile (Q3) and the first quartile (Q1) of the sample observation value, and $RMSE$ is the root mean square error.

The PLSR [39] model is a stoichiometric statistical model, which can solve the multicollinearity problem among independent variables, realizing data dimensionality reduction, information synthesis, and screening. Full cross validation was used in the modeling process.

The RF [40] is a machine learning algorithm for classification and regression. Based on decision tree learning and a simple average algorithm, the RF selects N samples according to the number of nodes (M) in each binary tree and the bootstrap method to construct the decision tree and then uses unselected samples to predict each tree. Because the RF randomly selects features and variables, overfitting can be avoided.

The ERT [41] is an ensemble learning method based on decision trees. If there is an initial training set of size N, in the extreme random tree, each decision tree is trained based on the whole dataset, which ensures the utilization of training samples and reduces the final prediction bias to a certain extent.

The RR [42] is an improved least squares method, which provides good results in ill-conditioned data processing and feature information extraction. It is also a new quantitative spectral analysis method.

## 3. Results

### 3.1. Descriptive Statistics of Measured Soil Attributes

According to the statistics of the properties of the six salinized soil types, the range of the EC values in the CSK was the largest, from 0.1 to 8.8 ms/cm, whereas for SSK the range of the EC values was the smallest, though relatively stable, at 1~3 ms/cm (Figure 2). The pH value of the TSN was the highest, followed by the HSK. The soil EC and pH levels did not change regularly, and a small EC had a higher pH (Table 2). The SSK had the highest SOM, whereas the TSN had the lowest. The SMC of the FSK was the highest, followed by the CSK, whereas that of the HSN was the lowest.

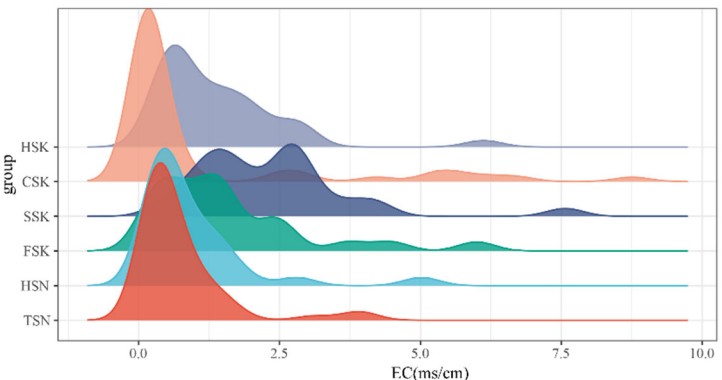

**Figure 2.** Numerical distribution of the sample points of different soil types. HSK: Haplic Solonchaks, SSK: Stagnic Solonchaks, CSK: Calcic Sonlonchaks, FSK: Fluvic Solonchaks, HSN: Haplic Sonlontzs, and TSN: Takyr Solonetzs.

**Table 2.** Summary statistics of the measured soil attributes of different soil types.

| Soil Types | EC Mean (Min–Max) | SD | pH Mean (Min–Max) | SD | SMC Mean (Min–Max) | SD | SOM Mean (Min–Max) | SD | Soil Texture (0–30) |
|---|---|---|---|---|---|---|---|---|---|
| HSK | 1.3 (0.2–6.1) | 1.2 | 8.3 (7.6–9.6) | 0.5 | 12.8 (1.1–24.6) | 6.5 | 13.3 (2.6–34.8) | 7.8 | Silt loam |
| CSK | 1.1 (0.1–8.8) | 2.1 | 8.5 (7.3–9.1) | 0.4 | 12.5 (2.1–26.0) | 7.4 | 7.6 (1.5–34.6) | 5.4 | Loam |
| SSK | 2.4 (0.4–7.6) | 1.5 | 7.9 (7.6–8.7) | 0.3 | 16.9 (2.4–26.1) | 6.4 | 25.2 (8.2–44.7) | 6.8 | Clay loam |
| FSK | 1.6 (0.1–6) | 1.4 | 8.0 (7.5–8.5) | 0.2 | 22.6 (5.4–39.8) | 6.4 | 20.4 (7.8–28.2) | 5.0 | Silty clay loam |
| HSN | 0.9 (0.1–5.0) | 1.0 | 8.3 (7.8–9.0) | 0.3 | 19.2 (11.0–24.11) | 2.9 | 17.3 (8.8–28.6) | 4.5 | Clay loam |
| TSN | 0.7 (0.2–3.9) | 0.8 | 8.7 (7.9–9.9) | 0.4 | 16.2 (1.4–35.9) | 5.6 | 12.4 (1.1–23.6) | 5.8 | Clay loam |

SOM is soil organic matter (g/kg); SMC is soil moisture content (%).

### 3.2. Hyperspectral Characteristics of Different Types of Salinized Soils

The pattern of the spectral curves tended to be consistent across the different soil types (Figure 3). Across the whole spectrum, i.e., 400–2400 nm, the spectral reflectance at 400–650 nm increased fastest; at 650–1400 nm, it increased steadily, and it fluctuated at 2100–2400 nm. The water absorption valleys were around 1400, 1950, and 2200 nm, with the most obvious one at 1900 nm.

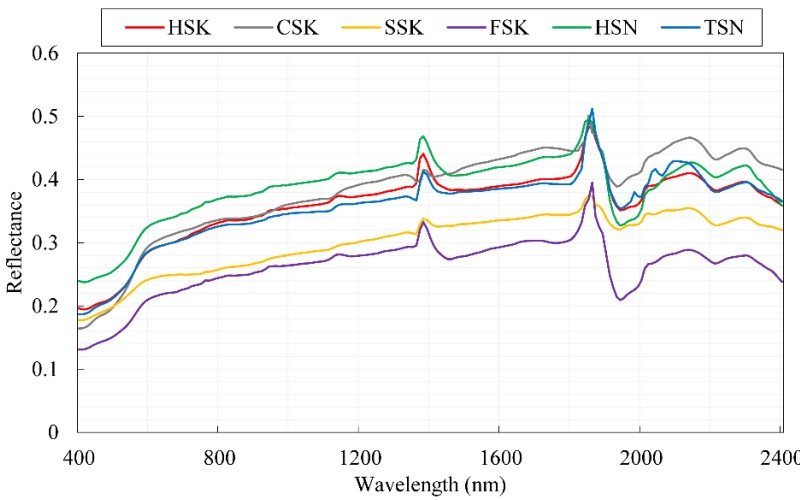

**Figure 3.** Mean spectral reflectance of various salinized soil types.

At 400–1400 nm, HSN's reflectance was the highest, FSK's reflectance was the lowest, and the reflectance of the HSK and TSN had the least difference. At 800–1400 nm, every soil type had a distinct reflectance, in the following order HSN > CSK > HSK > TSN > SSK > FSK. As the reflectance of the CSK was greater than that of the HSN after 1400 nm, it can be used to distinguish between the various salinized soil types. Regarding the shape of the curve, the slope of the reflectance curve of the CSK was the highest of all the soils at 400–700 nm, whereas the slopes of the reflectance curves of the other soils were roughly the same. The absorption depth and area of the water absorption characteristic zones were stronger for the FSK and HSN, at almost 1400 nm and 1900 nm, respectively, than for the other salinized soil types, and the absorption intensity of the SSK was the weakest. Throughout the whole spectrum, the FSK had the lowest reflectance, and the HSK and TSN spectra curves had similar patterns.

The spectral curves of the six salinized soils with different degrees of salinization continuously increased in the visible band (Figure 4), and the higher the salinization degree, the higher the reflectance. The increase in the reflectance of the HSK with the increase in salinization degree was at 400–650 nm, but at 650–1400 nm, the reflectance of the non-salinized, moderately salinized, strongly salinized, and extremely salinized soils were almost coincident, and after 1400 nm, the curves changed irregularly. The CSK's spectral reflectance increased with the increase in the wavelength over the whole spectrum. The reflectance of the moderately salinized soil was the highest, and that of the slightly salinized soil was the lowest, whereas the reflectance of the non-salinized and extremely salinized soils was similar at 400–1300 nm. After 1400 nm, the reflectance followed the order of moderately > non > extremely > slightly. The spectral reflectance of the SSK showed regular, gentle, and consistent changes at various degrees of salinization except for the moderate and strongly salinized soils. At 1400 and 1900 nm, the spectral reflectance of the moderate and extremely salinized soils of the SSK showed peaks, whereas the non-salinized, slightly salinized, and strongly salinized soils showed absorption valleys. At 1200–1900 nm, the difference between the hyperspectral reflectance curves of the different salinization degrees was the largest, and it was easy to distinguish the hyperspectral reflectance curves of soils with different EC values. The reflectance of the various salinization degrees of the FSK were close to each other with no regularity. Among the reflectance of different salinization degrees of the HSN, the non-salinized and slightly salinized showed regular changes, whereas the reflectance of the moderately and extremely salinized were similar at 400 nm–1300 nm, and the extremely salinized reflectance fluctuated greatly after 1400 nm. The absorption valley of the HSN soils was obvious and close to 1900 nm. The reflectance of the different salinization degrees of the TSN showed regular changes be-

tween 400 nm and 1900 nm, and the higher the EC value of the soil, the stronger the reflectance. However, the spectral difference between the slightly and moderately salinized soils was small.

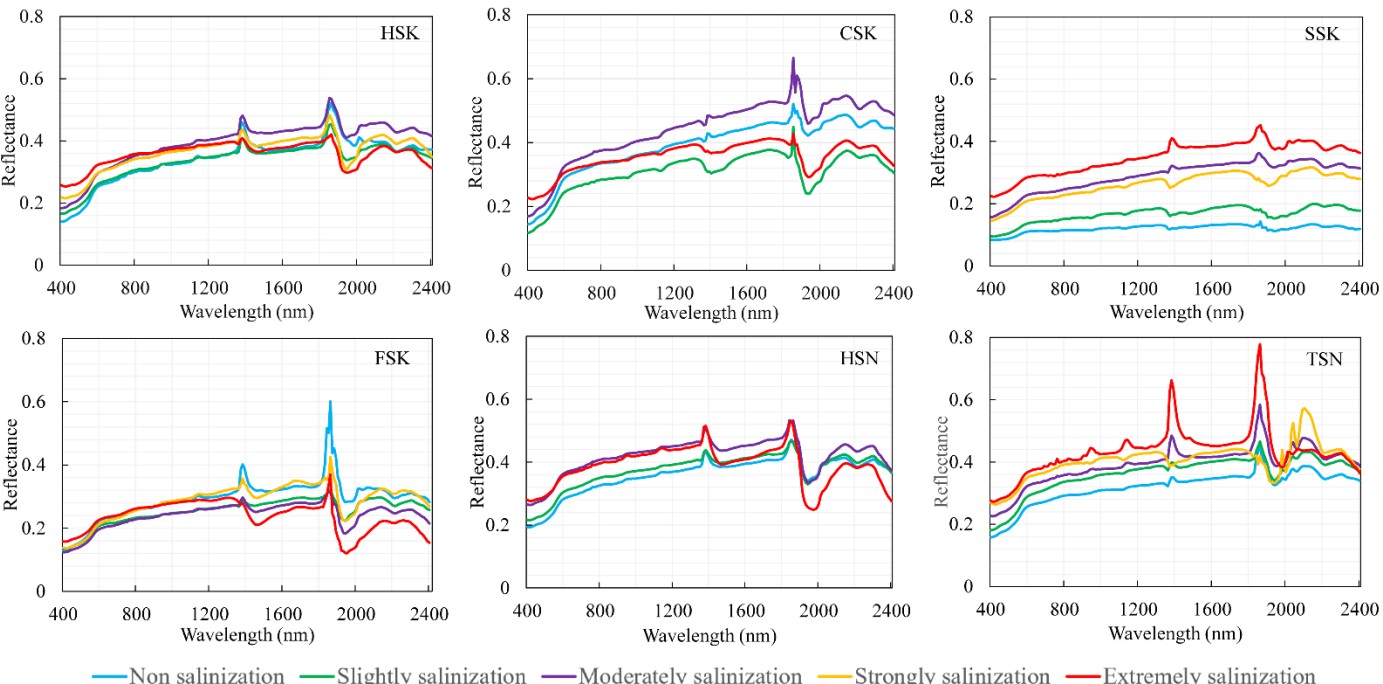

**Figure 4.** Spectral reflectance of the hyperspectral data depending on different salinization degree.

### 3.3. Correlation Coefficient between EC and Reflectance of Salinized Soil Types

The correlation between the spectrum and the EC of all the salinized soils of the different types was analyzed (Figure 5). The PCC between the reflectivity and conductivity of the HSN was almost unchanged at 400–1300 nm, but after 1300 nm, the fluctuation became larger. The HSN had an obvious "valley" shape at around 1400 nm and 2000 nm, followed by the SSK and FSK, for which the valleys were weak. Hardly any "valley" shape could be seen for the TSN in the whole band range. The EC and reflectance of the SSK and TSN were positively correlated in the whole band, and the correlation decreased over the spectrum. The EC values and reflectance of the SAS were negatively correlated as a whole, and the correlation between the EC values and the reflectance in the CSK, HSN, and HSK decreased over the whole spectrum (the absolute value decreased initially and then increased). The correlation became negative for the CSK, HSK, and HSN at close to 1100 nm, 1300 nm, and 1900 nm, respectively. The correlation between the EC values and the reflectance of these three soil types fluctuated after 1900 nm, with the most fluctuations observed in the HSN. In terms of the strength of the correlation, the correlation between the EC values and the reflectivity in the SSK was the strongest at 400–800 nm, followed by the TSN at 1000–2200 nm. The TSN had the largest correlation, whereas the HSK had the smallest. The correlation between the whole range of EC values and the reflectance was TSN > SSK > FSK > HSN > CSK > HSK. The SSK had the strongest positive correlation (0.70) with the reflectance at 400 nm, whereas the CSK had the strongest negative correlation (−0.45) with the reflectance at 1990 nm.

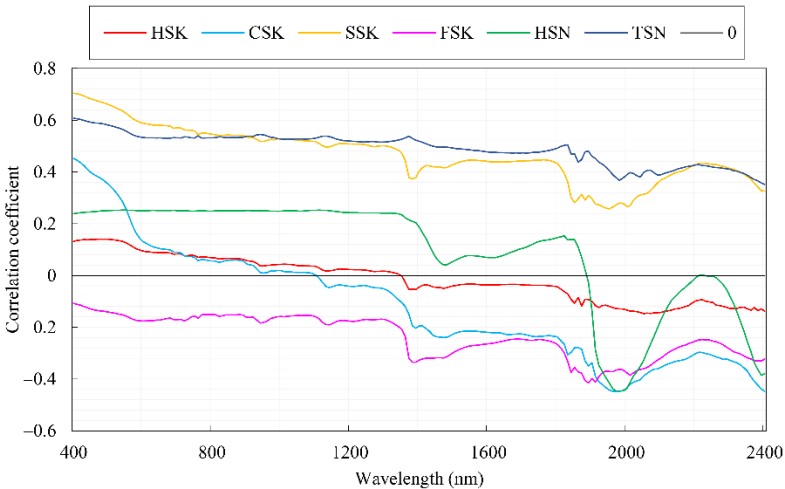

**Figure 5.** Correlation coefficient between the EC values and the reflectance spectra of various salinized soil types.

### 3.4. Relationship between Soil EC and Spectral Parameters

The soil EC values and the salinity index had a significant correlation (Figure 6). The HSK–NDI, CSK–RI, SSK–DI, FSK–RI, HSN–RI, and TSN–NDI provided the best results, with a maximum absolute PCC of 0.9651, 0.7751, 0.8072, 0.8459, 0.8731, and 0.7412, respectively. The correlation between the HSK–NDI and the EC values was the strongest, and its explicit expression was $[(R_{1600\,nm} - R_{1410\,nm})/(R_{1600\,nm} + R_{1410\,nm})]$. Overall, the best bands of the FSK–DI and TSN–DI were concentrated, whereas the other best bands were scattered, mostly in the form of grids and dots.

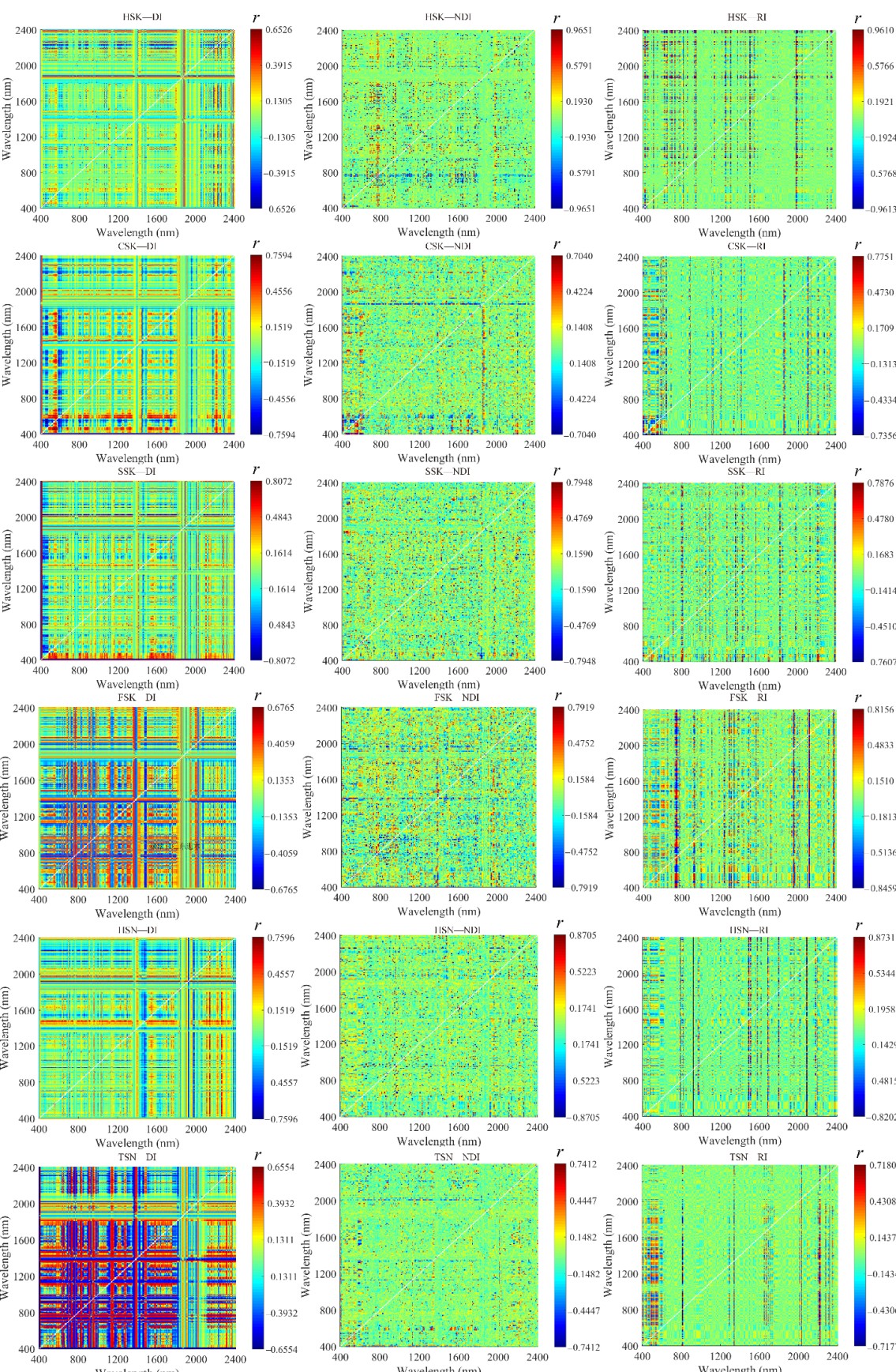

**Figure 6.** Two-dimensional correlation coefficients between the EC values and the salinity index under two derivative orders (The x and y axis represent the wavelength 400–2400 nm. The right-side color bar indicates the color of the PCC values. The colors dark red and dark blue represent a

relatively high PCC (red for positive and blue for negative) between the measured EC and the band combinations).

### 3.5. Optimal Factor Selection for Soil EC Inversion

In the case of the six kinds of saline soils, the number of independent variables selected from 21 independent variables using the VIP were 7, 8, 9, 4, 8, and 9, respectively, as shown in Figure 7.

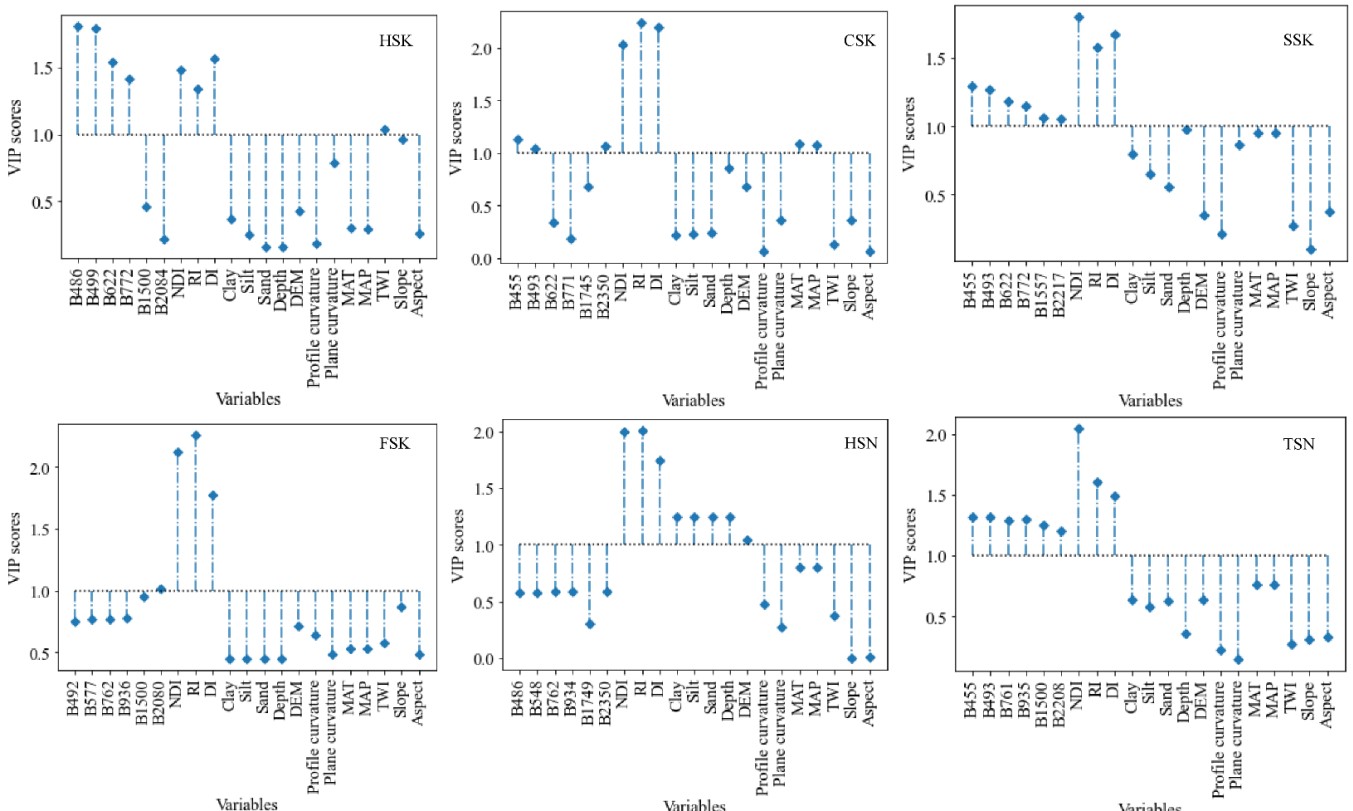

**Figure 7.** Variable importance projection (VIP) analysis between soil EC and variables.

### 3.6. Model Establishment

Taking the parameters obtained from the VIP screening as independent variables and the soil EC values as dependent variables, the PLSR, RF, ERT, and RR were used to build quantitative inversion models of the soil EC. The optimal hyperparameters of the model are shown in Table 3. Of the four models, the ERT had the best and most stable performance, followed by the RF, PLSR, and RR (Table 4). The most effective inversion model differed according to the saline soil type. The RF was the best model for the HSK and FSK, whereas the ERT was the best for the CSK, SSK, HSN, and TSN. Of the six saline soils, the results of the four inversion models for the HSN were the most stable with an $R^2$ ranging from 0.80 to 0.94 and an RMSE averaging 0.32. The difference in the effect of the models was the largest when applied to the TSN, ranging from the PLSR with an $R^2 = 0.50$ to the ERT ($R^2 = 0.92$), whereas for the HSK all the models showed similar results with an $R^2$ at 0.52–0.61. The ERT model performed best on the CSK ($R^2 = 0.99$, RMSE = 0.18, and RPIQ = 6.38). Overall, the ERT had the best prediction ability, with an average RMSE of 0.37, which was the lowest of the four models. Moreover, the training time for ERT was shorter than for the RF. Therefore, it can be concluded that the ERT has a good ability to predict the soil EC value.

**Table 3.** Optimal hyperparameters of the machine learning methods based on hyperspectral data.

| Category | Method | Optimal Hyperparameters |
|---|---|---|
| HSK | PLSR | n_components = 1 |
| | RF | n_estimators = 42, max_depth = 2, max_features = 2, random_state = 1 |
| | ERT | n_estimators = 17, max_depth = 2, random_state = 1 |
| | RR | alpha = 0.01 |
| CSK | PLSR | n_components = 10 |
| | RF | n_estimators = 45, max_depth = 2, max_features = 6, random_state = 1 |
| | ERT | n_estimators = 15, max_depth = 4, random_state = 1 |
| | RR | alpha = 7 |
| SSK | PLSR | n_components = 2 |
| | RF | n_estimators = 47, max_depth = 2, max_features = 6, random_state = 1 |
| | ERT | n_estimators = 24, max_depth = 4, random_state = 1 |
| | RR | alpha = 0.5 |
| FSK | PLSR | n_components = 1 |
| | RF | n_estimators = 19, max_depth = 6, max_features = 4, random_state = 1 |
| | ERT | n_estimators = 2, max_depth = 5, random_state = 1 |
| | RR | alpha = 0.5 |
| HSN | PLSR | n_components = 4 |
| | RF | n_estimators = 19, max_depth = 8, max_features = 6, random_state = 1 |
| | ERT | n_estimators = 3, max_depth = 4, random_state = 1 |
| | RR | alpha = 10 |
| TSN | PLSR | n_components = 1 |
| | RF | n_estimators = 5, max_depth = 4, max_features = 8, random_state = 1 |
| | ERT | n_estimators = 18, max_depth = 3, random_state = 1 |
| | RR | alpha = 100 |

**Table 4.** Inversion model of soil EC value based on hyperspectral data.

| Method | HSK | | | CSK | | | SSK | | |
|---|---|---|---|---|---|---|---|---|---|
| | $R^2$ | RMSE | RPIQ | $R^2$ | RMSE | RPIQ | $R^2$ | RMSE | RPIQ |
| PLSR | 0.56 | 0.74 | 1.80 | 0.84 | 0.82 | 1.62 | 0.78 | 0.65 | 2.05 |
| RF | **0.61** | **0.69** | **1.93** | 0.93 | 0.54 | 2.46 | 0.79 | 0.62 | 2.15 |
| ERT | 0.60 | 0.71 | 1.87 | **0.99** | **0.18** | **6.38** | **0.88** | **0.46** | **2.89** |
| RR | 0.52 | 0.78 | 1.71 | 0.75 | 1.04 | 1.28 | 0.72 | 0.72 | 2.85 |
| Method | FSK | | | HSN | | | TSN | | |
| | $R^2$ | RMSE | RPIQ | $R^2$ | RMSE | RPIQ | $R^2$ | RMSE | RPIQ |
| PLSR | 0.81 | 0.60 | 2.22 | 0.89 | 0.33 | 4.03 | 0.50 | 0.57 | 2.33 |
| RF | **0.93** | **0.36** | **3.69** | 0.91 | 0.29 | 4.59 | 0.89 | 0.27 | 4.93 |
| ERT | 0.90 | 0.43 | 3.09 | **0.94** | **0.24** | **5.54** | **0.92** | **0.22** | **6.05** |
| RR | 0.77 | 0.66 | 2.01 | 0.80 | 0.43 | 3.09 | 0.73 | 0.42 | 3.17 |

The scatter plots of the soil salinity measured and predicted by the best model, i.e., the ERT, showed that the CSK–ERT model performed optimally in linking the independent variables with the soil EC value (Figure 8).

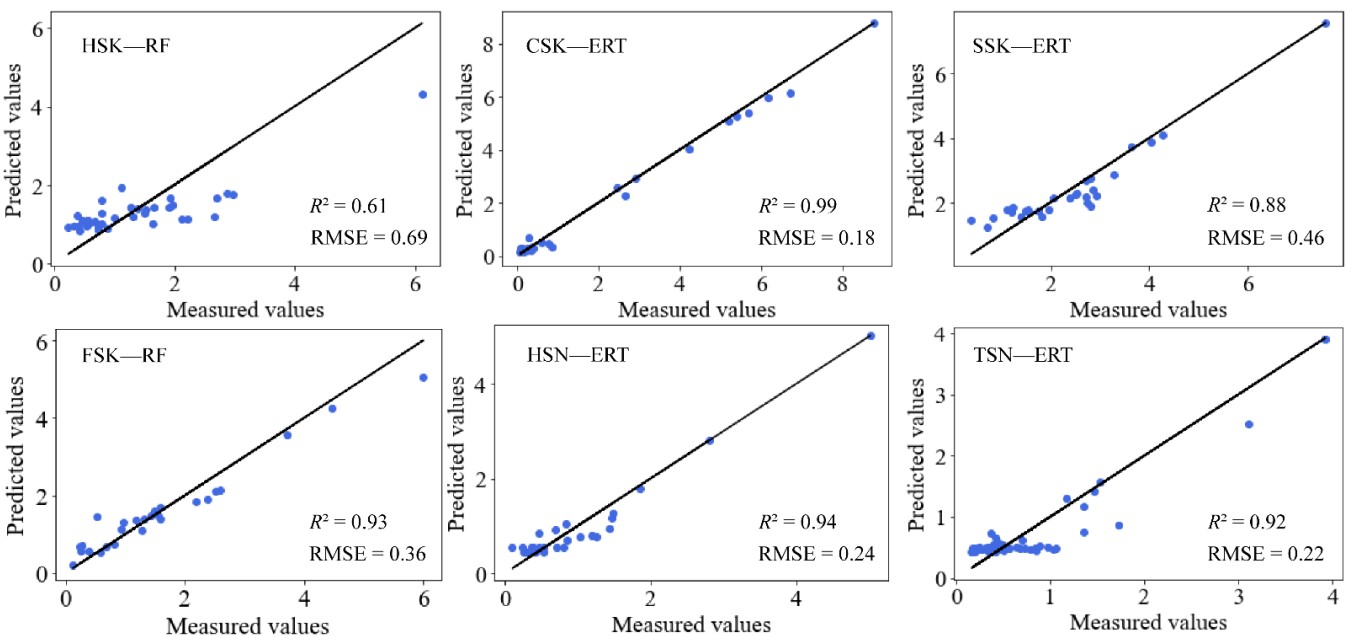

**Figure 8.** Scatter plots of the measured EC values and the best model-predicted values for different saline soil types.

## 4. Discussion

### 4.1. Spectral Characteristics of Different Types of Soil

The pattern of the reflection curves of the different saline soil types presented similar morphological characteristics (Figure 3) at 1400 nm and 1900 nm, with an obvious ever present water vapor absorption band, caused by the double or combined vibration frequency of the soil water molecules [43]. However, the size of the hyperspectral reflectance differed between the soils, because of the differences in soil formation conditions and the parent materials, i.e., the soil mineralogy factors that control the spectral characteristics. With the exception of the SSK, the size of the reflectance did not regularly change with the degree of salinity (Figure 4) and differed from the values reported from other locations, such as the Ebinur Lake oasis [18] and Zhenlai County in China [44] and the Urmia Lake in Iran [45]. This can be attributed to the relatively high pH but low EC values in some parts of our study area. An example of this can be seen in Table 2: $pH_{min}$ = 7.3 and the corresponding $EC_{min}$ = 0.1, which directly led to the irregular changes in the spectral characteristic curve of the salinized soil at different degrees of salinity. Furthermore, we analyzed various soil types, whereas other studies only considered one soil type.

The different correlation properties between the spectral reflectance and the EC values of different saline soil types (Figure 5) can be attributed to differences in the salinization mechanisms and the spatial heterogeneity of the salinity [46]. The higher maximum correlation coefficient between the two-dimensional spectral index and the EC values compared with that between the original band reflectance and the EC after the second derivative (Figure 6) indicated that the spectral index reduced the influence of noise to a large extent, took account of the remote sensing mechanism, and could dynamically extract soil EC spectral information [47,48].

### 4.2. Inversion of Soil Salinity Based on VIP Feature Screening

Soil salinity levels are controlled by various environmental factors. Therefore, the robustness of the model can be improved by removing potentially irrelevant environmental variables [49]. Among the 21 environmental covariates initially considered in this study, the two-dimensional spectral index (RI, NDI, and DI) had the highest selection frequency of the six saline soil types under the VIP selection (Figure 7), similar to the findings of

previous studies [50]. After the two-dimensional spectral index, the hyperspectral bands had the highest selection frequency among the soil types, whereas the elevation factors, climate variables, and soil texture were included to a lesser extent. Despite the importance of climate and topography as non-negligible soil formation factors and in determining the direction and rate of solute migration in soils, in addition to controlling the soil moisture regime and water temperature which directly control solute distribution in soils [51,52], of the VIP screening model factors, the terrain factors TWI and DEM were selected as modeling factors for the HSK and HSN, respectively, whereas the terrain factors in the other soil types failed to pass the screening. This indicates that climate and topography have little influence on the reflectance of the various salinity degrees of a specific soil type. Nevertheless, all our sampled soils were cultivated on relatively flat topography, with low heterogeneity. The relative importance of climate variables was even lower than that of terrain, with the exceptions of the MAT and MAP for the CSK. However, none of the climate variables passed the screening, because the CSK is in the south of the Hetao plain, where the rainfall is far greater than it is in the other northern locations (Figure 1).

The machine learning model was significantly more accurate than the linear model (Table 4). In general, the ERT model performed best, followed by the RF and PLSR, whereas the RR model was the least effective, comparable with previous research results [53]. The reason for this is that the machine learning algorithm (random forest) introduced random attribute selection during model training and extracted data based on randomness and differences, which improved the accuracy of decision making [54]. The PLSR model could correct the collinearity problem [55]. However, the PLSR fitting also reduced the dimension of the data, leading to the loss of point data information to a certain extent. Thus, the inversion accuracy decreased. As a multivariate linear regression model, the RR also achieved good results in the inversion of some of the saline soils (CSK, FSK, and HSN), but the effect was not as good as that of the PLSR model. Some studies have pointed out that as a biased estimation method, the RR was more consistent with the actual regression process [56].

### 4.3. Model Uncertainty Analysis

The key to effectively predicting soil salinity (EC) using spectroscopy (VIS-NIR) depends on the proper selection of soil and environmental characteristics and the model. The data, spectral covariates, and the model are the most common sources of uncertainty [57]. The uncertainty of the spectral covariates is mainly attributable to the different effects of soil organic matter and water on the spectrum. The salinization mechanisms in our study resulted from the infiltration of exogenous water, topography, inappropriate cultivation management with regard to climate, and geology, i.e., the saline parent materials. In particular, the perennial irrigation without drainage and the annual introduction of a large amount of irrigation water from the Ningxia section of the Yellow River with a water salinity level of 0.5 g/L [58] both increased the salinity of the soils. Furthermore, irrigation water side seepage causes the adjacent lowland groundwater level to rise, resulting in secondary soil salinization. Therefore, the distribution of salt in the study area neither changes according to depth nor is constant over time. The micro-topography of farmland soil and the adsorption characteristics of the soil components change the location of the soil salt deposition. The uneven distribution of the soil samples in the study area leads to an uneven density of the soil samples with different degrees of salinization.

The texture, soil depth, water content, and organic matter of the different types of saline soil were inconsistent (Table 2), which affected the soil spectrum. The spatial scale of the predictor variables has a significant impact on the prediction accuracy [59]. The soil texture affects the absorption, reflection, and scattering characteristics of visible near-infrared spectra from the physical structure of particle composition and the chemical characteristics of clay particles [60,61], which further affects the model inversion effect. Studies have shown that the higher the clay content in the soil, the higher the EC value, and this plays a significant role in the model [23]. The larger the range from bedrock to surface, the

larger the soil volume and the lower the salt content under the influence of natural and human factors [62]. Different types of saline–alkaline soil have different soil depths. In this study, the different types of salinized soils were not collected at the same scale because of the different sample sites, and this also led to inconsistent model accuracy.

Our study did not consider the influence of water and land surface temperature, because soil salinity is closely related to the groundwater level. This is also recognizable from the correlation coefficient of −0.603 to −0.705 between the groundwater depth and salt content of the cultivated topsoil (0–20 cm) in Ningxia [63]. Thermodynamic factors such as heat capacity and the coefficient of thermal conductivity are also likely to affect salt deposition and distribution, but this effect should be more reflected in the time scale of soil salinization dynamics. The management of cultivated land soil is a fundamental element of the process of secondary salinization. The distances of irrigation and drainage infrastructure, the use of chemical fertilizer, and planting patterns are all reported as variables affecting salinity [64,65]. In addition, this study only used the measured hyperspectral data for research and did not use the data from multispectral remote sensing. In future research, we will combine the multispectral remote sensing data with the measured hyperspectral data for salinization inversion research.

## 5. Conclusions

To study the differences in the hyperspectral characteristics of various saline–alkaline soil types and establish high precision quantitative inversion models, we selected the Hetao plain on the upper reaches of the Yellow River, where the soils have different degrees of salinization resulting from various salinization processes. The patterns of the spectra curves of different salinized soil types were generally the same, but the size of the hyperspectral reflectance differed. Up to 1400 nm, the Haplic Sonlontzs (HSN) had the highest reflectance, followed by the Calcic Sonlonchaks (CSK). The Haplic Solonchaks (HSK) and Takyr Solonetzs (TSN) showed similar reflectance, and the Fluvic Solonchaks (FSK) had the lowest reflectance. The spectral curves of the soils with degrees of salinization in the HSK, CSK, HSN, and TSN increased with the increase in the salinity at 400–650 nm, but after 650 nm, the reflectance was irregular. The reflectance of the Stagnic Solonchaks (SSK) soils with degrees of salinization were quite different from each other, with regular changes except for the moderate and strongly salinized soils. The reflectance of the FSK soils with different degrees of salinization showed similar changes without regularity. The heterogeneity of the various salinized soil types led to inconsistent correlation properties between the soils. In the whole band, the reflectance of the SSK and TSN were positively correlated with the EC values, but the FSK was negatively correlated, and the correlation of the reflectance of the HSK, CSK, and HSN with the EC values changed from positive to negative. Based on the variable projection importance (VIP), different characteristic factors were selected for the various salinized soil types. The two-dimensional spectral index (RI, DI, and NDI) and characteristic bands were the most selected factors, whereas the topographic variables and climatic variables were less sensitive to the EC. Of the four modeling methods applied, the model performance was extremely randomized trees (ERT) > random forest (RF) > partial least squares regression (PLSR) > ridge regression (RR). The most effective inversion model for the HSK and FSK was the RF, and for the CSK, SSK, HSN, and TSN, it was the ERT. Of the models, the CSK–ERT was the most effective ($R^2 = 0.99$, RMSE = 0.18, and RPIQ = 6.38). This study provides a reference for the inversion of the soil EC values of cultivated land, which has a wide range of salinity, and lays a foundation for large-scale monitoring of soil salinization using remote sensing.

**Author Contributions:** Writing—Original Draft and Visualization, P.J.; Data curation, P.J., J.Z., and K.J.; Validation, P.J. and X.Z.; Formal analysis, W.H., D.Y., Y.H., and X.Z.; Supervision, J.Z. and X.Z.; Review, editing, and revision, K.Z. and X.Z.; Funding acquisition, J.Z., K.J., K.Z., and X.Z. All authors have read and agreed to the published version of the manuscript.

**Funding:** This work was supported by the National Natural Science Foundation of China (Grant numbers 41877109; 42050410320; 42067003; 42061047); the Jiangsu Specially-Appointed Professor Project, China (Grant number R2020T29); the Key R&D Project of Ningxia, China (Grant number 2021BEG03002); the National Key R&D Program of China (Grant number 2021YFD1900602); and the Open Fund of Tsinghua University-Ningxia Yinchuan Joint Research Institute of Water Networking and Digital Water Control (SKLHSE–2022–IOW11).

**Data Availability Statement:** Not applicable.

**Conflicts of Interest:** The authors declare no conflicts of interest.

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
