# Peer review of "Inversion of Different Cultivated Soil Types’ Salinity Using Hyperspectral Data and Machine Learning"

_remotesensing, doi:10.3390/rs14225639_

Round 1

Reviewer 1 Report

This paper is an interesting study on an important topic. The authors have gathered enough data and have performed quite a few steps to analyze them. However, there are a few shortcomings. Many English mistakes to be found throughout, the paper needs a very detailed proof-reading process to ameliorate this. I have noted a few of them below, but there are many more. I regret to also state that the report suffers from lack of scientific details at various steps that are most important. A few steps of the analysis seem incoherent and may not be followed through. A comparison between e.g. the raw reflectance hyperspectral values, and the use of the suggested band indices and/or the ancillary data (topographical, climatic, etc.) is missing. For all these reasons I suggest a major review of the project.
Moderate / major remarks:
1. Figure 1, you have not explained how you selected the sampling locations
2. You should explicitly state why you opted to record surface spectra with a bare optic fiber, and not e.g. a) use the contact probe with its own illumination source, b) measure the spectra in the lab, etc.
3. Lines 139 to 140,  please be more explicit what step 1 entails
4. Line 142, please be more explicit about the parameters of the Savitzky-Golay filter
5. Line 145 to 148, you have said that you did resampling at 10nm, why are these numbers given with 1nm resolution?
6. Section 2.4.2 please report more rigorously how R2 was calculated as there are some conflicting opinions
7. Section 2.4.2 please report what the 5-fold CV entails (random partitions into folds?), if you used an independent test set (or just CV), how you optimized the models’ hyperparameters and what were the search spaces, etc.
8. Table 2 is not summary statistics (e.g. no std, variance, median, quartiles etc.) but only the range of the values
9. I would have liked to see a cross-correlation between the soil properties reported on Table 2, e.g. between EC, pH and SOM.
10. Figure 3 is a bit peculiar, is this after all the pre-processing steps? What are those peaks at around the water absorption bands? Is it because of the noise introduced by the atmospheric water? Add a grid line and minor ticks to aid with the readability of the exact wavelengths. Make the legend lines thicker to aid with the readability of the colors. Same notes for Figures 4 and 5.
11. Section 3.3, this is calculated across all salinization degrees, correct? State so explicitly.
12. Why is section 3.5 stated before the development of the PLSR model? Isn’t the order confusing? I think you should re-phrase things a bit here and describe more rigorously how the model was established. I also don’t understand again why we see things like B2217 when the data was resampled with 10nm resolution.
13. Table 3, I am missing the number of optimal latent variables for PLSR.
14. Table 4, I suggest you also include the RPIQ metric and use boldface to denote the optimal results. I guess this is in the independent test set? Please report more rigorously.
15. Figure 8 and in general, I guess this is after 5-fold CV. This is most spectacular because usually it is very difficult to predict the large values that are handful when the most abundant values are in the lower range. Take a close look at the highest values of TSN-ERT, SSK-ERT, etc. where the prediction is almost 100% accurate, even when at the low values you have some errors in prediction. Usually when predicting such properties that have large skewness the problem is so difficult that sometimes we use the sqrt or log transforms. Why did you believe you have such great results? Does this perhaps indicate an issue with the sampling or the split in the folds? I would investigate more closely here.
16. Sections 3.5 and 3.6; are these results using only the initial hyperspectral values (reflectance)? Is there a comparison between using hyperspectral only and hyperspectral + ancillary data (e.g. topography, climatic)? If not, why provide these data and report on them on Figure 7? Why calculate the different indices if you are not going to use them? Or are you using them for model establishment? This is most confusing. You need to make everything 100% clearer for the reader.
17. A more critical view of the manuscript is missing. What are the drawbacks? What are some suggestions for future work?
Minor remarks:
* In abstract “and have measured hyperspectral” is not specific, is this in situ? Lab? Vis-NIR-SWIR? SWIR only? Etc.
* Line 26 “for inversion …” rephrase
* “A certain number of” line 128, this is not so scientific
* Line 142, capitalize G
* Line 145, define PCC
* Line 400, “the band had

Author Response

October 18th, 2022

Dear Reviewer,

Thank you so much for the review of our manuscript (remotesensing - 1951099) entitled “Inversion of Cultivated Soil Salinity on Hyperspectral and Machine Learning Based on Soil Types".

The detailed responses comments are included in the following pages.

Yours sincerely

Xiaoning Zhao

The School of Geographical Sciences

Nanjing University of Information Science and Technology

Ningliu Road 219, Nanjing, China

Tel: +86 17351789670

Email: jasminezxnsx@msn.com

Response to Reviewer 1 Comments

This paper is an interesting study on an important topic. The authors have gathered enough data and have performed quite a few steps to analyze them. However, there are a few shortcomings. Many English mistakes to be found throughout, the paper needs a very detailed proof-reading process to ameliorate this. I have noted a few of them below, but there are many more. I regret to also state that the report suffers from lack of scientific details at various steps that are most important. A few steps of the analysis seem incoherent and may not be followed through. A comparison between e.g. the raw reflectance hyperspectral values, and the use of the suggested band indices and/or the ancillary data (topographical, climatic, etc.) is missing. For all these reasons I suggest a major review of the project.

Response: Thank you so much for your affirmation of our research. We solve the questions step by step.

  1. Many English mistakes to be found throughout

Response: we have already revised the whole manuscript through English service from MDPI. See the certificate as followed:

  1. lack of scientific details at various steps that are most important. A comparison between e.g. the raw reflectance hyperspectral values, and the use of the suggested band indices and/or the ancillary data (topographical, climatic, etc.) is missing.

Response: It has been supplemented in the paper (model building method, evaluation method, data descriptive statistics), To eliminate instrument noise and environmental background interference, the edge bands with excessive noise (350-399 and 2401-2500 nm) were removed. Among the selected model modeling factors, the original bands with the largest correlation with EC in the range of blue (455-492 nm), green (492-577 nm), red (622-770 nm), near infrared (770-1050 nm), swir1 (1500-1750 nm) and swir2 (2080-2350 nm) were used. The rest of the spectral operations were performed after 10 nm resampling. The salinity index and auxiliary data in this paper are the selected model modeling factors, but some variables may have little correlation with conductivity, so the variables with large correlation with conductivity were selected for modeling by VIP screening method before modeling.

Point 1: Figure 1, you have not explained how you selected the sampling locations.

Response 1: Considering the factors such as soil surface characteristics, pH conditions, soil types and land use patterns in the study area. We select the 6 sampling locations, where the main 6 saline and alkaline soil types in Hetao plain. To explain clear for the readers, we add Haplic Solonchaks (HSK), Stagnic Solonchaks (SSK), Takyr Solonetzs (TSN), Haplic Sonlontzs (HSN), Fluvic Solonchaks (FSK) and Calsic Sonlonchaks (CSK) in material and methods.

Point 2: You should explicitly state why you opted to record surface spectra with a bare optic fiber, and not e.g. a) use the contact probe with its own illumination source, b) measure the spectra in the lab, etc.

Response 2: Compared with remote sensing, in situ remote sensing mainly refers to the method that can quickly obtain soil characteristics within 1m depth near the surface soil. Some studies have shown that the field measured spectrum can well retrieve the soil attributes [1-3]. The contact probe with its own illumination source must be maintained good and in the field in-situ measurement of soil spectrum, it is necessary to avoid objects with higher hardness to avoid breaking the mirror surface [4]. The field condition is complicated. The field soil is not damaged and the field spectrum can better reflect the measured information of soil samples compared the sampled soil in lab. Therefore, the field measured spectra are selected to participate in the study.

Point 3: Lines 139 to 140, please be more explicit what step 1 entails.

Response 3: The abnormal spectral curve removal, breakpoint correction and the measured field spectral data were redone in ViewSpec Pro software (A click view graph was used to delete the abnormal curve. The ASD spectrometer has three sensors, which have varying responsivity under different environmental function temperatures and warm-up times. Different optical fibers collect spectra of samples at different locations, and the splice correction function in the software was required to correct the data). It has been supplemented in the text.

Point 4: Line 142, please be more explicit about the parameters of the Savitzky-Golay filter

Response 4: Savitzky-Golay (Polynomial order is 2, Number of smoothing points is 9), It has been supplemented in the text.

Point 5: Line 145 to 148, you have said that you did resampling at 10nm, why are these numbers given with 1nm resolution?

Response 5: In order to give consideration to the smoothing of spectral curve and spectral characteristics, the spectral data of 400~2400 nm are resampled at 10 nm intervals, and the spectral bands consisting of 201 wavebands are obtained. In order to more accurately select the band with the largest correlation with conductivity within the blue (455-492 nm), green (492-577 nm), red (622-770 nm), near-infrared (770-1050 nm), swir1 (1500-1750 nm) and swir2 (2080-2350 nm) range to participate in EC modeling, the original hyperspectral band (before resampling) is selected.

Point 6: Section 2.4.2 please report more rigorously how R2 was calculated as there are some conflicting opinions.

Response 6: It has been supplemented in the text. We explain it in detail and we also add RMSE and RPIQ to evaluation the data better in material and methods part as followed:

                              (1)

Where yi and  is the observed value and predicted value of the test sample,  is the average of sample observations, and n is the number of predicted samples.

                      (2)

Where  I is the predicted value of the sample, yi is the measured value.

                                 (3)

Where IQ is the difference between the third quartile (Q3) and the first quartile (Q1) of the sample observation value, and RMSE is the root mean square error.

Point 7: Section 2.4.2 please report what the 5-fold CV entails (random partitions into folds?), if you used an independent test set (or just CV), how you optimized the models’ hyperparameters and what were the search spaces, etc. 

Response 7: Grid SearchCV is selected as the super parameter selection in this paper. The effect of Grid Search method is affected by the division of initial data. A five fold cross validation method is introduced to reduce the chance of parameter adjustment results. K-fold cross validation can ensure the stability of the training model and make the model have good effect on each subset. A five fold cross validation method is used to allocate training sets and validation sets. Step: divide all data sets into five parts, select four as the training set each time, and the rest as the verification set. Train the model several times until each part has been divided into the training set and verification set; Calculate the mean square error of each verification set, and average the mean square error values of 5 times to obtain the final MSE. We add the sentence “The model search spaces are: PLSR param_grid = {'n_components': range (1, 20)}, RF: 'n_estimators': range (2, 50, 1), 'max_features': [2, 4, 6, 8], 'max_depth': range (2, 15, 2); ERT: 'n_estimators': range (1, 30, 1), 'max_depth': range (2, 15, 1), RR: ridge = RidgeCV (alphas=[0.01, 0.1, 0.5, 1, 5, 7, 10, 30,100]).” to show how to optimized the models’ hyperparameters and the search space.

Point 8: Table 2 is not summary statistics (e.g. no std, variance, median, quartiles etc.) but only the range of the values. 

Response 8: Yes, you are right. The statistical data have been supplemented in the paper as followed.

Table 2. Summary statistics of measured soil attributes at different soil types

Soil types

EC

pH

SMC

SOM

Soil texture

(0-30)

Mean(min-max)

SD

Mean(min-max)

SD

Mean(min-max)

SD

Mean(min-max)

SD

HSK

1.3(0.2-6.1)

1.2

8.3(7.6-9.6)

0.5

12.8(1.1-24.6)

6.5

13.3(2.6-34.8)

7.8

Silt loam

CSK

1.1(0.1-8.8)

2.1

8.5(7.3-9.1)

0.4

12.5(2.1-26.0)

7.4

7.6(1.5-34.6)

5.4

Loam

SSK

2.4(0.4-7.6)

1.5

7.9(7.6-8.7)

0.3

16.9(2.4-26.1)

6.4

25.2(8.2-44.7)

6.8

Clay loam

FSK

1.6(0.1-6)

1.4

8.0(7.5-8.5)

0.2

22.6(5.4-39.8)

6.4

20.4(7.8-28.2)

5.0

Silty clay loam

HSN

0.9(0.1-5.0)

1.0

8.3(7.8-9.0)

0.3

19.2(11.0-24.11)

2.9

17.3(8.8-28.6)

4.5

Clay loam

TSN

0.7(0.2-3.9)

0.8

8.7(7.9-9.9)

0.4

16.2(1.4-35.9)

5.6

12.4(1.1-23.6)

5.8

Clay loam

Point 9: I would have liked to see a cross-correlation between the soil properties reported on Table 2, e.g. between EC, pH and SOM. 

Response 9: We conducted correlation analysis on soil attributes as followed:

Point 10: Figure 3 is a bit peculiar, is this after all the pre-processing steps? What are those peaks at around the water absorption bands? Is it because of the noise introduced by the atmospheric water? Add a grid line and minor ticks to aid with the readability of the exact wavelengths. Make the legend lines thicker to aid with the readability of the colors. Same notes for Figures 4 and 5.

Response 10: Yes, this is after all the pre-processing steps; Figure 3 shows the soil spectral curve obtained by averaging different types of saline soil. There are peaks near 1400, 1950 and 2200 nm, the appearance of water vapor absorption peak is caused by double frequency or combined frequency of soil water molecule vibration, which was also related to clay mineralogy; The figure has been modified and grid lines have been added and the legend lines were thicker as shown below (Fig. 3, 4, 5):

Figure 3. Mean spectral reflectance of various salinized soil types.

Figure 4. Spectral reflectance of hyperspectral depending on salinization degree.

Figure 5. Correlation coefficient between EC values and the reflectance spectra of various salinized soil types

Point 11: Section 3.3, this is calculated across all salinization degrees, correct? State so explicitly.

Response 11: Yes, you are right. The correlation diagram in 3.3 shows the correlation between reflectivity and conductivity of all samples of different types of saline soil. It has been supplemented in the text.

Point 12: Why is section 3.5 stated before the development of the PLSR model? Isn’t the order confusing? I think you should re-phrase things a bit here and describe more rigorously how the model was established. I also don’t understand again why we see things like B2217 when the data was resampled with 10nm resolution.

Response 12: In order to find the hyperspectral sensitive bands in different band ranges (blue (455-492 nm), green (492-577 nm), red (622-770 nm), near-infrared (770-1050 nm), swir1 (1500-1750 nm) and swir2 (2080-2350 nm)) more accurately and participate in modeling, the sensitive bands are screened in the original hyperspectral bands. In addition to the selected sensitive bands, all spectral calculations in this paper are performed at a resampling resolution of 10 nm. Because all models are built on the basis of VIP screening characteristic bands, that is, the VIP screening characteristic factors are used to participate in modeling, so the VIP results need to be described before the model is built. The band factors in the model are selected on the basis of the original hyperspectrum, so there will be bands that are not multiples of 10.

Point 13: Table 3, I am missing the number of optimal latent variables for PLSR.

Response 13: Yes, you are right. It has been supplemented in the text as followed.

Table 3. Optimal hyperparameters of machine learning methods based on hyperspec-tral.

Category

Method

Optimal hyperparameters

HSK

PLSR

n_components=1

RF

n_estimators= 42, max_depth= 2, max_features= 2, random_state= 1

ERT

n_estimators= 17, max_depth= 2, random_state= 1

RR

alpha=0.01

CSK

PLSR

n_components=10

RF

n_estimators= 45, max_depth= 2, max_features= 6, random_state= 1

ERT

n_estimators=15, max_depth=4, random_state= 1

RR

alpha=7

SSK

PLSR

n_components=2

RF

n_estimators= 47, max_depth= 2, max_features= 6, random_state=1

ERT

n_estimators= 24, max_depth= 4, random_state= 1

RR

alpha= 0.5

FSK

PLSR

n_components=1

RF

n_estimators= 19, max_depth= 6, max_features= 4, random_state= 1

ERT

n_estimators= 2, max_depth= 5, random_state= 1

RR

alpha= 0.5

HSN

PLSR

n_components=4

RF

n_estimators= 19, max_depth= 8, max_features= 6, random_state= 1

ERT

n_estimators= 3, max_depth= 4, random_state= 1

RR

alpha= 10

TSN

PLSR

n_components=1

RF

n_estimators= 5, max_depth= 4, max_features= 8, random_state= 1

ERT

n_estimators= 18, max_depth= 3, random_state= 1

RR

alpha=100

Point 14: Table 4, I suggest you also include the RPIQ metric and use boldface to denote the optimal results. I guess this is in the independent test set? Please report more rigorously.

Response 14: Yes, you are right. It has been supplemented in the text as followed.

Table 4. Inversion model of soil EC value based on hyperspectral.

Method

HSK

CSK

SSK

R2

RMSE

RPIQ

R2

RMSE

RPIQ

R2

RMSE

RPIQ

PLSR

0.56

0.74

1.80

0.84

0.82

1.62

0.78

0.65

2.05

RF

0.61

0.69

1.93

0.93

0.54

2.46

0.79

0.62

2.15

ERT

0.60

0.71

1.87

0.99

0.18

6.38

0.88

0.46

2.89

RR

0.52

0.78

1.71

0.75

1.04

1.28

0.72

0.72

2.85

Method

FSK

HSN

TSN

R2

RMSE

RPIQ

R2

RMSE

RPIQ

R2

RMSE

RPIQ

PLSR

0.81

0.60

2.22

0.89

0.33

4.03

0.50

0.57

2.33

RF

0.93

0.36

3.69

0.91

0.29

4.59

0.89

0.27

4.93

ERT

0.90

0.43

3.09

0.94

0.24

5.54

0.92

0.22

6.05

RR

0.77

0.66

2.01

0.80

0.43

3.09

0.73

0.42

3.17

Point 15: Figure 8 and in general, I guess this is after 5-fold CV. This is most spectacular because usually it is very difficult to predict the large values that are handful when the most abundant values are in the lower range. Take a close look at the highest values of TSN-ERT, SSK-ERT, etc. where the prediction is almost 100% accurate, even when at the low values you have some errors in prediction. Usually when predicting such properties that have large skewness the problem is so difficult that sometimes we use the sqrt or log transforms. Why did you believe you have such great results? Does this perhaps indicate an issue with the sampling or the split in the folds? I would investigate more closely here.

Response 15: It is after 5-fold CV. The organic matter content here is very low [5], so the impact of organic matter on reflectivity is small. The impact of EC on reflectivity is big. The models used in this paper are random forest and extremely randomized trees. All belong to Bagging method, which is mainly based on Bootstrapping to randomly sample n samples from the sample training set, repeatedly sample i times, get i training sets and train the model respectively, and then get the final result by voting or averaging. The importance of each basic learner is the same. It is not easy to fall into over fitting, and has strong adaptability to data sets. Therefore, the prediction effect of different soil types of saline soil in this paper is better.

Point 16: Sections 3.5 and 3.6; are these results using only the initial hyperspectral values (reflectance)? Is there a comparison between using hyperspectral only and hyperspectral + ancillary data (e.g. topography, climatic)? If not, why provide these data and report on them on Figure 7? Why calculate the different indices if you are not going to use them? Or are you using them for model establishment? This is most confusing. You need to make everything 100% clearer for the reader.

Response 16: The original hyperspectral reflectance was used for the model characteristic bands in Sections 3.5 and 3.6, and the spectral reflectance after 10 nm resampling was used for the calculation of salinity index. Hyperspectral and hyperspectral+auxiliary data were not modeled and compared in this paper. Because some environmental variables may not provide information to predict the characteristics of the target soil, they may be redundant or highly relevant. VIP feature selection method can filter feature variables. After determining the relevant variables for each soil EC, these selected environmental variables were then used for modeling analysis of each soil EC. Figure 7 VIP Filtering for All Variables.

Point 17: A more critical view of the manuscript is missing. What are the drawbacks? What are some suggestions for future work?

Response 17: Yes, you are right. This part of the content has been rewritten in discussion.

In this study, different types of saline soil do not follow the same scale, therefore the saline soil hyperspectral inversion should be considered the soil types in the futures. In addition, this paper only uses the measured hyperspectral data for research, and does not use the multispectral remote sensing. In the later research, we will combine the multispectral remote sensing data with the measured hyperspectral data for salinization inversion research in the follow-up study.

Point 18: In abstract “and have measured hyperspectral” is not specific, is this in situ? Lab? Vis-NIR-SWIR? SWIR only? Etc.

Response 18: “and have measured situ hyperspectral, pH value and electrical conductivity (EC) of totally 231 soil samples. It has been supplemented and modified in the text.

Point 19: Line 26 “for inversion …” rephrase

Response 19: Various models of partial least square regression (PLSR), random forest (RF), extremely randomized trees (ERT) and ridge regression (RR) were used for inversion under saline soil types.

Point 20: “A certain number of” line 128, this is not so scientific

Response 20: Yes, you are right. Soil spectra were measured after harvest in each sampling site. It has been supplemented and modified in the text.

Point 21: Line 142, capitalize G

Response 21: Yes, you are right. Savitzky-Golay. It has been supplemented and modified in the text.

Point 22: Line 145, define PCC

Response 22: Yes, you are right. Pearson’s correlation coefficient(PCC). It has been supplemented and modified in the text.

Point 23: Line 400, “the band had

Response 23: Yes, you are right. It has been supplemented and modified in the text. the hyperspectral bands had the highest selection frequency among the soil types.

References

  • Peng, J.; Wang, J.Q.; Xiang, H.Y.; et al. Comparative study on hyperspec-tral inversion accuracy of soil salt content and electrical conductivity. Spectroscpy and Spectral Analysis, 2014, 34(2), 510–514.
  • Vaudour, E.; Gilliot, J.M.; Bel, L.; et al. Regional prediction of soil organic carbon content over temperate croplands using visible near-infrared airborne hyperspectral imagery and synchronous field spectra, International Journal of Applied Earth Observation and Geoinformation, 2016, 49, 24–38.
  • Zhang, J.H.; Jia, P.P.; Sun, Y.; et al. Prediction of salinity ion content in different soil layers based on hyperspectral data. Transactions of the Chinese Society of Agricultural Engineering (Transactions of the CSAE), 2019, 35(12): 106–115.
  • Shi, Z. et al. Principle and method of hyperspectral remote sensing of soil surface. Science Press. Beijing, 2014.5.
  • Shang, T.H.; Mao, H.X.; Zhang, J.H., et al. Hyperspectral estimation of soil organic matter content in Yinchuan plain, China based on PCA sensitive band screening and SVM modeling. Chinese Journal of Ecology, 2021, 40(12): 4128–4136.

Reviewer 2 Report

This paper provides a reference for distinguishing and monitoring various salinization types based on hyperspectral images. The paper is well organized. However, the major concern of the reviewer is that the motivation of the paper should be better clarified. Please see the suggestions below.

‘Soil salinization is one of the main causes of global desertification and soil degradation. Although many researches have studied the hyperspectral inversion of soil salinity with machine learning, a few were based on soil types. ’ Why it is important to estimate soil salinity according to soil types and what are the major challenges to doing so?

‘This study provides ….provides a foundation for the accurate monitoring of salinized soil by multi-spectral remote sensing.’ Does it mean hyperspectral remote sensing?

The accuracy of the methods is evaluated with R2. Since the ground truth is available. Why not evaluate the methods with accuracy? 

The evaluated methods are all traditional methods. It is suggested to compare some deep-learning-based methods since they normally perform better than traditional methods. 

Author Response

October 18th, 2022

Dear Reviewer,

Thank you so much for the review of our manuscript (remotesensing - 1951099) entitled “Inversion of Cultivated Soil Salinity on Hyperspectral and Machine Learning Based on Soil Types".

The detailed responses comments are included in the following pages.

Yours sincerely

Xiaoning Zhao

The School of Geographical Sciences

Nanjing University of Information Science and Technology

Ningliu Road 219, Nanjing, China

Tel: +86 17351789670

Email: jasminezxnsx@msn.com

Response to Reviewer 2 Comments

This paper provides a reference for distinguishing and monitoring various salinization types based on hyperspectral images. The paper is well organized. However, the major concern of the reviewer is that the motivation of the paper should be better clarified. Please see the suggestions below.

Response: the motivation of the paper is emphasized in the paper with sentences in “introduction” and “abstract “.

Point 1: ‘Soil salinization is one of the main causes of global desertification and soil degradation. Although many researches have studied the hyperspectral inversion of soil salinity with machine learning, a few were based on soil types.’ Why it is important to estimate soil salinity according to soil types and what are the major challenges to doing so?

Response 1: The formation factors of saline soil of different soil types are different, and the soil texture, mechanical composition, organic matter and other aspects are different. The treatment of saline soil needs to be implemented accurately, and different soil types determine the management measures for crops and agriculture, so it is necessary to understand different types of saline soil.

Challenges: Perhaps the hyperspectral retrieval of salt is not affected by the soil type, or the spectral retrieval may not be related to the soil type. There has no research whether can spectrum be used to determine the salinization level and different soil types? In terms of technical conditions, hyperspectral inversion has not been combined with images. It is still a question to combine it with remote sensing for later promotion for providing field management according to different salt level to larger scale.

Point 2: ‘This study provides ….provides a foundation for the accurate monitoring of salinized soil by multi-spectral remote sensing.’ Does it mean hyperspectral remote sensing?

Response 2: Multispectral remote sensing includes multispectral images and hyperspectral images. In this paper, field in-situ soil hyperspectral data are measured, VIP method was used to screen sensitive modeling factors for different types of saline soil, and soil EC inversion models were established. R2, RMSE and RPIQ were used to evaluate the accuracy of different models, and good results were obtained. Situ hyperspectral model can be extended to the multispectral remote sensing.

Point 3: The accuracy of the methods is evaluated with R2. Since the ground truth is available. Why not evaluate the methods with accuracy? 

Response 3: We add R2, RMSE and RPIQ for accuracy as followed:

                              (1)

Where yi and  is the observed value and predicted value of the test sample,  is the average of sample observations, and n is the number of predicted samples.

                      (2)

Where  I is the predicted value of the sample, yi is the measured value.

                                 (3)

Where IQ is the difference between the third quartile (Q3) and the first quartile (Q1) of the sample observation value, and RMSE is the root mean square error.

Method

HSK

CSK

SSK

R2

RMSE

RPIQ

R2

RMSE

RPIQ

R2

RMSE

RPIQ

PLSR

0.56

0.74

1.80

0.84

0.82

1.62

0.78

0.65

2.05

RF

0.61

0.69

1.93

0.93

0.54

2.46

0.79

0.62

2.15

ERT

0.60

0.71

1.87

0.99

0.18

6.38

0.88

0.46

2.89

RR

0.52

0.78

1.71

0.75

1.04

1.28

0.72

0.72

2.85

Method

FSK

HSN

TSN

R2

RMSE

RPIQ

R2

RMSE

RPIQ

R2

RMSE

RPIQ

PLSR

0.81

0.60

2.22

0.89

0.33

4.03

0.50

0.57

2.33

RF

0.93

0.36

3.69

0.91

0.29

4.59

0.89

0.27

4.93

ERT

0.90

0.43

3.09

0.94

0.24

5.54

0.92

0.22

6.05

RR

0.77

0.66

2.01

0.80

0.43

3.09

0.73

0.42

3.17

Point 4: The evaluated methods are all traditional methods. It is suggested to compare some deep-learning-based methods since they normally perform better than traditional methods. 

Response 4: The performance of the traditional machine learning model is already good and the CSK-ERT has the best performance (R2=0.99, RMSE=0.18, RPIQ=6.38). In the subsequent research, we will consider the combination of multispectral remote sensing and near earth remote sensing, and use the depth learning and traditional methods to invert the soil conductivity in several different places.

Reviewer 3 Report

This paper shows the results of retrieving soil electrical conductivity (EC) using ground-measured hyperspectral data. This study emphasizes that the best retrieval method varies for different soil types. This study was well-designed and provided good examples of how soil types affect the spectrum of salinized soils and its consequent challenge to find the best retrieval method suitable for most soil types. I think that this paper provides contributions to this topic and could be published after some refinements.

1. The two figures in Figure 2 provided the same information, so I suggest keeping only one figure. The curves in Figure 2b are not necessary.

2. The contents in section 3.7 help little to the discussion of this research and could be omitted.

3. I suggest the author compare the retrieval of EC by considering soil types with not considering soil types. Suppose the author could retrieve EC using all the data collected in the field not considering soil types and compare with the retrieval results of considering soil types. In that case, it’ll be more convincing whether soil type plays a vital role in retrieving EC using the hyperspectral of salinized soils.

4. The English expression should be improved; some long sentences should be rephrased for better understanding.

Author Response

October 18th, 2022

Dear Reviewer,

Thank you so much for the review of our manuscript (remotesensing - 1951099) entitled “Inversion of Cultivated Soil Salinity on Hyperspectral and Machine Learning Based on Soil Types".

The detailed responses comments are included in the following pages.

Yours sincerely

Xiaoning Zhao

The School of Geographical Sciences

Nanjing University of Information Science and Technology

Ningliu Road 219, Nanjing, China

Tel: +86 17351789670

Email: jasminezxnsx@msn.com

Response to Reviewer 3 Comments

This paper shows the results of retrieving soil electrical conductivity (EC) using ground-measured hyperspectral data. This study emphasizes that the best retrieval method varies for different soil types. This study was well-designed and provided good examples of how soil types affect the spectrum of salinized soils and its consequent challenge to find the best retrieval method suitable for most soil types. I think that this paper provides contributions to this topic and could be published after some refinements.

Point 1: The two figures in Figure 2 provided the same information, so I suggest keeping only one figure. The curves in Figure 2b are not necessary.

Response 1: Figure 2b has been deleted from the text.

Point 2: The contents in section 3.7 help little to the discussion of this research and could be omitted.

Response 2: Section 3.7 has been deleted in the text.

Point 3: I suggest the author compare the retrieval of EC by considering soil types with not considering soil types. Suppose the author could retrieve EC using all the data collected in the field not considering soil types and compare with the retrieval results of considering soil types. In that case, it’ll be more convincing whether soil type plays a vital role in retrieving EC using the hyperspectral of salinized soils.

Response 3: All saline alkali soil data were modeled using the same method, and the results are as follows:

evaluating indicator

PLSR

RF

ERT

RR

R2

0.63

0.90

0.72

0.58

RMSE

1.29

0.66

1.12

1.38

RPIQ

1.01

1.97

1.16

0.94

Model Performance of all saline alkali soil data was not better than that of different soil types. At most important, the formation factors of saline soil of different soil types are different, and the soil texture, mechanical composition, organic matter and other aspects are different. Moreover, this paper mainly studies the inversion of electrical conductivity of different types of salinized soil because the treatment of saline soil needs to be implemented accurately, and different soil types determine the management measures for crops and agriculture.

Point 4: The English expression should be improved; some long sentences should be rephrased for better understanding. 

Response 4: The English expression has been revised.

Round 2

Reviewer 2 Report

The authors have addressed the concerns of this reviewer.